# ILP-FORMER: Solving Integer Linear Programming with Sequence to Multi-Label Learning

**Shufeng Kong**[1,2]        **Caihua Liu**[3*]        **Carla Gomes**[2]

[1]School of Software Engineering, Sun Yat-sen University, Zhuhai, Guangdong, China
[2]Computer Science Dept., Cornell University, Ithaca, NY, USA
[3]School of Artificial Intelligence, Guilin University of Electronic Technology, Guilin, Guangxi, China

## Abstract

Integer Linear Programming (ILP) is an essential class of combinatorial optimization problems (COPs). Its inherent NP-hardness has fostered considerable efforts towards the development of heuristic strategies. An emerging approach involves leveraging data-driven methods to automatically learn these heuristics. For example, using deep (reinforcement) learning to recurrently reoptimize an initial solution with Large Neighborhood Search (LNS) has demonstrated exceptional performance across numerous applications. A pivotal challenge within LNS lies in identifying an optimal subset of variables for reoptimization at each stage. Existing methods typically learn a policy to select a subset, either by maintaining a fixed cardinality or by decomposing the subset into independent binary decisions for each variable. However, such strategies overlook the modeling of LNS's sequential processes and fail to explore the correlations inherent in variable selection. To overcome these shortcomings, we introduce ILP-FORMER, an innovative model that reimagines policy learning as a sequence-to-multi-label classification (MLC) problem. Our approach uniquely integrates a causal transformer encoder to capture the sequential nature of LNS. Additionally, we employ an MLC decoder with contrastive learning to exploit the correlations in variable selection. Our extensive experiments confirm that ILP-FORMER delivers state-of-the-art anytime performance on several ILP benchmarks. Furthermore, ILP-FORMER exhibits impressive generalization capabilities when dealing with larger problem instances.

## 1 INTRODUCTION

ILP has found applications in production planning Mula et al. [2006], scheduling Ku and Beck [2016], scientific discovery Chen et al. [2021a], and telecommunications networks Gollowitzer and Ljubić [2011], among many others. It is well-known that ILP is NP-complete Karp [1972] and many efforts have been devoted to designing effective heuristics to find near-optimal solutions Taha [2014]. Historically, such algorithms were designed largely manually, requiring a careful understanding of the underlying structure within specific classes of optimization problems.

Due to the recent success of deep learning (DL) and reinforcement learning (RL), there has been an increasing interest in automatically learning heuristics for COPs from training data Bengio et al. [2021]. Existing works often leverage machine learning (ML) to output solutions directly from input instances, configure hyperparameters of COP algorithms, or learn a local decision policy for search frameworks such as branch&bound (B&B), local branching (LB), or LNS. Among them, we are particularly interested in learning to iteratively reoptimize an initial solution with LNS Wu et al. [2021], Song et al. [2020], Sonnerat et al. [2021], Nair et al. [2020a]. These approaches are attractive because we can leverage existing state-of-the-art commercial ILP solvers such as Gurobi or SCIP as a generic black-box subroutine and thus benefits from the cutting-edge technologies of such commercial ILP solvers.

In this paper, we focus on boosting the performance of LNS, though our method can also be applied to boost the performance of other local search algorithms such as LB. A key challenge of LNS is to select a promising variable subset to reoptimize based on the current solution. Since the selection choice is combinatorial, finding an optimal subset is also computationally hard. Song et al. Song et al. [2020] learn to select fixed, predefined variable subsets with imitation learning and RL. Wu et al. Wu et al. [2021] learn to select arbitrary variable subsets with RL by factorizing the selection of a variable subset into elementary selections on each

---

*Correspondence to: Caihua Liu <Caihua.Liu@guet.edu.cn>

variable separately. Similarly, Sonnerat et al. Sonnerat et al. [2021] learn to predict the probability of selecting a variable independently of other variables using imitation learning and Nair et al. Nair et al. [2020a] use RL to learn a policy that selects one variable at a time. Recently, Huang et al. Huang et al. [2023] adopt contrastive learning for a better embedding of ILPs. Nevertheless, all of these works miss modeling the sequential processes of LNS and also do not exploit correlations of variable selection. To address these limitations, we propose to model the policy learning as a sequence to a multi-label classification problem, which jointly models the selection of variables as well as the sequential processes of LNS.

Our contributions are threefold: (1) we give a new angle of sequence to multi-label classification for learning an effective local decision policy for LNS; (2) we materialize this idea by providing a novel model to seamlessly integrate a customized decision transformer encoder to model the sequential processes of LNS and an MLC decoder with contrastive learning to exploit correlations of variable selection; (3) we conduct extensive experiments on various benchmarks and the results show that our model significantly outperforms state-of-the-art baselines

## 2 OTHER RELATED WORK

**Learning to Optimize.** Recently, there has been an increasing interest in applying ML to learn solving COPs. Broadly speaking, there are three categories of learning to optimize algorithms: (1) Learning to predict solutions from inputs. Larsen et al. [2018] train a deep neural network (DNN) to predict the solution of a stochastic load planning problem. Nair et al. [2020b] propose neural diving to learn a DNN to generate multiple partial assignments for its integer variables, and the resulting smaller mixed integer programs (MIPs) for unassigned variables are solved with an off-the-shelf MIP solver to construct high-quality joint assignments. Joshi et al. [2019] learn a DNN by supervision to predict the probability of an edge to be in the traveling salesman problem (TSP) tour. A feasible tour is then generated by beam search. (2) Learning to configure COP algorithms. Liu et al. [2022] learn to configure the search neighborhood size of LB in each step by using RL. Deng et al. [2022b] integrate belief propagation (BP), gated recurrent units (GRUs), and graph attention networks (GATs) within the message-passing framework to reason about dynamic weights and damping factors for composing new BP messages. (3) Learning alongside COP algorithms. Nair et al. [2020a] learn a DNN to make variable selection decisions in B&B to bound the objective value gap with a small tree. Deep Bucket Elimination (DBE) Razeghi et al. [2021] uses DNNs to approximate the large bucket functions. Deng et al. [2022a] propose a pre-trained cost model which predicts the optimal cost of a given partially instantiated COP. The predicted cost is then used to construct heuristics for various COP algorithms such as LNS and B&B. Our work belongs to the third category.

**Primal Heuristics.** Numerous primal heuristic algorithms have been proposed to enhance the efficiency of solving ILPs Berthold [2013]. Primal heuristics span from simpler rounding heuristics Achterberg et al. [2012] to more computationally demanding diving and large neighborhood search (LNS) heuristics, such as Relaxation Induced Neighborhood Search (RINS) Danna et al. [2005]. LNS heuristics are improvement heuristics that solve auxiliary problems using the branch-and-bound technique. In contrast, learning-based LNS approaches can be regarded as primal heuristics automatically learned through machine learning. These approaches showcase significant potential by exploiting data-driven techniques, which ultimately result in improved performance and adaptability across a wide range of problem instances. This work is particularly interested in advancing the capabilities of learning-based LNS approaches.

## 3 PRELIMINARIES

### 3.1 INTEGER PROGRAM

An integer linear program (ILP) is a problem of optimizing a linear function over points in a polyhedral set: $\arg\min_x \{\mu^T x | Wx \leq b; x \geq 0; x \in \mathbb{Z}^n\}$, where $x \in \mathbb{Z}^n$ is a vector of $n$ decision variables; $\mu \in \mathbb{R}^n$ denotes the vector of objective coefficients; the incidence matrix $W \in \mathbb{R}^{m \times n}$ and vector $b \in \mathbb{R}^m$ together define $m$ linear constraints.

### 3.2 LNS AND ITS MARKOV DECISION PROCESS FORMULATION

Given an initial assignment of values to the decision variables in an ILP instance, LNS iteratively refines this assignment by selecting a subset of decision variables, relaxing their values, and solving a subproblem that aims to optimize the objective function while respecting the instance's constraints. LNS aims to explore a complex solution neighborhood and gradually improve its current solution until a certain termination condition is met Pisinger and Ropke [2010]. A key challenge of LNS is how to define a good solution neighborhood, namely, one needs to decide which variable subset to reoptimize given the current solution. Obviously, such a decision problem is combinatorial, and many works devote to constructing effective heuristics for it Ropke and Pisinger [2006], Perron et al. [2004], Dumez et al. [2021]. In this work, we are particularly interested in the recent trend of learning-based approaches, where data-driven methods are applied to learn the heuristics automatically Song et al. [2020], Wu et al. [2021], Sonnerat et al. [2021]. To this end, the LNS framework can be formulated as a *Markov Decision Process* (MDP) $(\mathcal{S}, \mathcal{A}, P, R)$:

- $\mathcal{S}$ is a set of states. A state $s_t \in S$ describes the current status of the LNS process in step $t$, which normally includes the static IP instance information (e.g., variables, constraints, and objectives) and the dynamic solving statistics (e.g., the incumbent solution);

- $\mathcal{A}$ is a set of all candidate variable subsets for reoptimization. A variable subset $a_t \in \mathcal{A}$ is also called an *action* of an agent that is executed in step $t$;

- $P(s_t, a_t)$ is the transition function to return the next state. Let $x_t$ be the solution with state $s_t$, a smaller sub-IP is first generated by keeping the values of non-selected variables in $x_t$ and reoptimizing the remainder, and then the next state $s_{t+1}$ is obtained by updating $s_t$ with the new solution to the sub-IP: $x_{t+1} = \arg\min_x \{\mu^T x | Wx \le b; x \ge 0; x \in \mathbb{Z}^n; x^i = x_t^i, \forall x^i \notin a^t\}$;

- $R(s_t, a_t)$ is the reward function to return the change of objective values, which is defined as $r_t = R(s_t, a_t) = \mu^T(x_t - x_{t+1})$. Let $T$ be the step limit, the *cumulative rewards* from step $t$ of an episode is defined as $R_t = \sum_{k=t}^{T} \gamma^{k-t} r_k$ with a discount factor $\gamma \in [0, 1]$.

A *policy* is a (potentially probabilistic) mapping $\pi : \mathcal{S} \to \mathcal{A}$. The goal of RL-based algorithms for solving ILPs is to find a policy function to maximize the expected cumulative reward $\mathbb{E}[R_1]$ over all episodes, i.e., the expected improvement over initial solutions. However, existing RL-based algorithms for IP solving train a policy by either temporal difference (TD) learning Sutton and Barto [2018], policy gradient Williams [1992], or behavior cloning Torabi et al. [2018], all of which miss modeling sequential processes of LNS explicitly. Furthermore, RL-based algorithms may suffer from various issues, such as the need for bootstrapping to propagate returns in TD-learning can cause stability problems, the discounting future rewards can induce undesirable short-sighted behaviors, policy gradient is known to be sample inefficient, and behavior cloning can suffer from cascading errors Kaelbling et al. [1996], Ross and Bagnell [2010], Chen et al. [2021b]. To circumvent these disadvantages, we propose to learn a policy with decision transformers, which seeks to benefit from modeling sequential processes of LNS and better generalization.

### 3.3 DECISION TRANSFORMER

Decision transformer (DT) Chen et al. [2021b] abstracts the decision-making process in RL as a sequence modeling problem and attempts to learn a return-conditioned state-action mapping. The return-conditionality means that given a history of return-state-action tokens, such that the first token represents the desired return at the current state, the DT predicts the action required to achieve this desired return. In this paper, we follow the convention of the original DT and define return, $g_t$, as the non-discounted rewards-to-go:

$g_t = \sum_t^T r_t$. DT takes as input a sequence of three-tokens: $(\langle g_{t-K}, s_{t-K}, a_{t-K} \rangle, \cdots, \langle g_t, s_t, a_t \rangle)$, where $K \le T$ is the context length. Each token is then encoded into an embedding and added by a positional encoding. Furthermore, let $(\langle z_{g_{t-K}}, z_{s_{t-K}}, z_{a_{t-K}} \rangle, \cdots, \langle z_{g_t}, z_{s_t}, z_{a_t} \rangle)$ be the corresponding sequence of embeddings, and this sequence of embeddings is fed into a causal transformer to produce another sequence of embeddings $(\langle z_{g_{t-K}}^h, z_{s_{t-K}}^h, z_{a_{t-K}}^h \rangle, \cdots, \langle z_{g_t}^h, z_{s_t}^h, z_{a_t}^h \rangle)$. A decoder takes as input $z_{s_t}^h$ and outputs $\hat{a}_t$. During training, a suitable loss function is applied to penalize the difference between the prediction $\hat{a}_t$ and label $a_t$. During inference, after specifying a target return based on desired performance and the environment starting state, DT generates actions autoregressively. The actions are executed and the target return is subtracted by the achieved rewards to obtain the next states. The process of generating actions and applying them to obtain the next return-to-go and state is repeated until episode termination.

## 4 SOLVING ILPS WITH SEQUENCE TO MULTI-LABEL CLASSIFICATION

Instead of learning to select fixed, predefined variable subsets Song et al. [2020], Wu et al. Wu et al. [2021] factorizes the combinatorial action space $\mathcal{A}$ into elementary actions on each dimension (i.e. variables), where $a_t^i \in \{1, 0\}$ denotes the elementary action of whether selecting $x^i$ for reoptimization in step $t$, and $a_t^i$ is 1 if $x^i$ is selected and 0 otherwise. Therefore, any action can be expressed as $a_t = \cup_{i=1}^n a_t^i$ and the action selection problem can be converted into $n$ separated binary classification problems. The policy for action selection is factorized by

$$\pi(a_t|s_t) = \prod_{i=1}^n \pi^i(a_t^i|s_t), \tag{1}$$

which expresses the probability of selecting an action as the product of probabilities of selecting its elements. However, such an action space factorization limits the class of policies that can be learned and it also fails to explore the correlations between elementary actions. To address these limitations, *we propose to model the policy learning as a sequence to multi-label classification problem, which jointly models the selection of multiple elementary actions as well as the sequential processes of LNS*, i.e.,

$$\pi(a_t|s_t) = p_\theta(a_t|h_\phi(Q(t, K))), \tag{2}$$

where $Q(t, K)$ denotes the function used to return the last $K$ sequence of return-state-action tokens from steps $t - K$ to $t$, namely, $(\langle g_{t-K}, s_{t-K}, a_{t-K} \rangle, \cdots, \langle g_t, s_t, \cdot \rangle)$; $h_\phi(\cdot)$ denotes the sequence encoder parameterized by $\phi$(NN); and the MLC decoder parameterized by $\theta$(NN) takes as input state embedding $z_{s_t}^h$ produced by $h_\phi$ and outputs action distribution $p_\theta(a_t|z_{s_t}^h)$. Effective implementations of the sequence encoder and MLC decoder are crucial to this work.

## 4.1 A NOVEL TRANSFORMER MODEL FOR SOLVING ILPS

In this section, we propose a novel model ILP-FORMER for the problem given in equation (2) based on the causal transformer and contrastive learning.

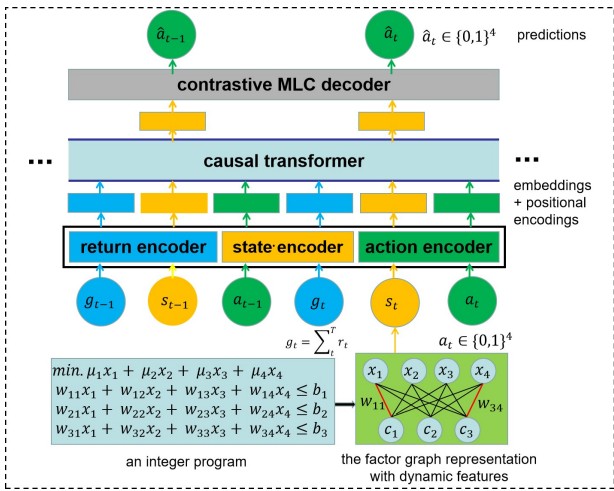

Figure 1: The architecture of our model ILP-FORMER: it consists of several token encoders to produce latent token embeddings, a causal transformer to capture dependence between token embeddings, and a contrastive MLC decoder to exploit correlations between label categories. Here we use a small IP with 4 variables and 3 constraints to show the full pipeline of our model. The problem is first translated into a factor graph $G$, and $G$ is associated with dynamic factor-node features that describe the states of MDP and are encoded into state embeddings in different steps. Similarly, returns $g_t$ and actions $a_t$ are also encoded into latent embeddings. We use a GCN as state encoder and two simple MLPs as return and action encoders respectively. Each token embedding is further added with its relative positional encoding. The sequence of embeddings is fed into the causal transformer to produce another sequence of embeddings. Finally, the contrastive MLC decoder takes as inputs the state embeddings and outputs action predictions.

### 4.1.1 Factor Graph Representation

An IP instance can be represented by a *factor graph* Gasse et al. [2019] which is a bipartite graph $\mathcal{G} = (\mathcal{V}, \mathcal{C}, \mathcal{E})$ consisting of variable-nodes $\mathcal{V} = \{v_1, \cdots, v_n\}$ and factor-nodes $\mathcal{C} = \{c_1, \cdots, c_m\}$. Variable nodes correspond to the variables and factor nodes correspond to the constraints in the IP. An edge $e_{ij} \in \mathcal{E}$ between $v_i$ and $c_j$ is established only if the $j$-th constraint contains the $i$-th variable. The variable nodes are associated with a feature matrix $V \in \mathbb{Z}^{n \times d_v}$, where $d_v$ is the number of features for each variable node. The features of each variable-node $v_i$ include two parts: (1) *static* features: a one-hot vector indicates the node type and the objective coefficient $\mu_i$ of $x_i$; (2) *dynamic* features: the

current solution of $x_i$ in step $t$ and the incumbent solution of $x_i$. Note that the dynamic features are used to describe the states of MDP in different steps. The factor nodes are also associated with a feature matrix $C \in \mathbb{Z}^{m \times d_c}$, where $d_c$ is the number of features for each factor node. The features of each factor-node $c_i$ only include static features: a one-hot vector indicates the node type and the value $b_i$ at the right-hand-side (RHS) of the $i$-th constraint. Finally, the weight matrix of edges is exactly the incidence matrix.

### 4.1.2 Model Architecture

Fig. 1 gives the overall architecture of our ILP-FORMER and it consists of a customized DT encoder and a contrastive MLC decoder. Our encoder is only composed of several customized token encoders and a causal transformer without the linear decoder of DT Chen et al. [2021b].

**Token Encoders**: Each token is first encoded into an embedding and added by a positional encoding. For return and action tokens, two simple multilayer perceptrons (MLPs) are used as return and action encoders respectively. Positional encodings are produced by another simple MLP which takes as input a single scalar $t$. Each state token $s_t$ is represented by a factor graph as introduced in section 4.1.1, and we use a graph convolutional network (GCN) Zhang et al. [2019] as the state encoder. A single graph convolution layer is detailed below

$$
\begin{aligned}
C^{(k+1)} &= C^{(k)} + \sigma\left(\text{LN}\left(WV^{(k)}H_v^{(k)}\right)\right), \\
V^{(k+1)} &= V^{(k)} + \sigma\left(\text{LN}\left(W^T C^{(k+1)} H_c^{(k)}\right)\right),
\end{aligned}
\tag{3}
$$

where $H_v^{(k)}, H_c^{(k)} \in \mathbb{R}^{d_h \times d_h}$ are trainable weight matrices in the $k$-th layer; $V^{(k)} \in \mathbb{R}^{n \times d_h}$ and $C^{(k)} \in \mathbb{R}^{m \times d_h}$ are embeddings for variable-nodes and factor-nodes respectively in the $k$-th layer; LN and $\sigma(\cdot)$ denote layer normalization and Tanh activation function respectively. The initial embeddings $V^{(0)}$ and $C^{(0)}$ are linear projections of the raw feature matrices $V$ and $C$ respectively. In this paper, all MLP encoders only have two layers, and the embeddings' dimensions $d_h$ are set to be 128; the GCN encoder consists of two convolution layers and a mean pooling layer.

**Causal Transformer**: Causal transformer Vaswani et al. [2017] is an architecture to efficiently model sequences that consist of stacked self-attention layers with residual connections. In our model, each layer receives a sequence of $L = 3K$ token embeddings $\{z_i\}_{i=1}^L$, and outputs $L$ embeddings $\{z_i^h\}_{i=1}^L$, preserving the input dimensions. Specifically, each token embedding $z_i$ is mapped to a key $z_i^k$, a query $z_i^q$, and a value $z_i^v$ via linear functions, and the output $z_i^h$ is given by

$$
z_i^h = \sum_{j=1}^i \alpha_{ij} z_j^v, \quad \alpha_{ij} = \frac{\exp(z_i^q \cdot z_j^k)}{\sum_{j'=1}^i \exp(z_i^q \cdot z_{j'}^k)}.
\tag{4}
$$

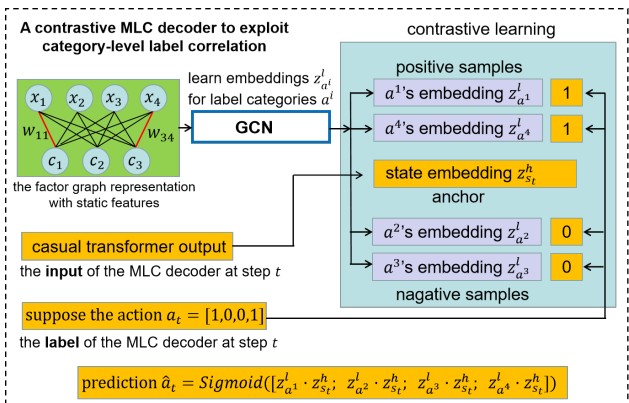

Figure 2: The architecture of our contrastive MLC decoder. Firstly, a GCN takes as input an IP instance and learns embeddings for label categories respectively, and labels within the same category share the same embedding. Secondly, we use the state embedding in step $t$ as an input feature whose inner products with label embeddings are used to produce prediction $\hat{a}_t$. Lastly, a contrastive loss is designed to pull together the feature embedding and positive label embeddings, while separating the feature embedding from the negative label embeddings.

In this work, we adopt the causal transformer GPT2 Radford et al. [2019] to learn and reason about sequences and we defer the other architecture details to the original paper.

**Contrastive MLC Decoder**: Recall that an elementary action $a_t^i \in \{0, 1\}$ (a.k.a. a label) denotes whether or not to select variable $x^i$ for reoptimization in step $t$. A label vector $a_t = \cup_{i=1}^n a_t^i \in \{0, 1\}^n$ denotes the selected action given state $s_t$. Different from eq. (1) which approximates $\pi(a_t|s_t)$ with $n$ separated binary classification problems, we propose to approximate $\pi(a_t|s_t)$ with an MLC decoder that finds a mapping from $z_{s_t}^h$ to $a_t$, where $z_{s_t}^h$ is a state embedding generated by the causal transformer and served as an input feature for our MLC decoder. *A key aspect of learning a policy with an MLC module is that we can exploit the correlations between elementary actions, which is missing in those existing ML-boosted IP solvers* Wu et al. [2021], Song et al. [2020], Sonnerat et al. [2021].

We propose to exploit category-level label correlation with contrastive learning based on the MLC model GMVAE Bai et al. [2022]. GMVAE assumes that the number of labels is fixed and label embeddings are shared across all samples. This is not applicable to our case since different instances may have a different number of decision variables, i.e., actions from different instances may have different cardinalities. Alternatively, we learn category-level label embeddings for each IP instance with a shared GCN, and the embeddings are only shared across samples within each instance. Fig. 2 gives the architecture of our MLC decoder. We denote $a^i$ the $i$-th category of labels $\{a_j^i\}_{j=t-K}^t$ collected from steps $[t-K, t]$. Our idea is as follows: (1) we learn an embedding

$z_{a^i}^l$ for each label category $a^i$ such that labels within the same category share the same embedding. Since the number of label categories is exactly the number of variables in an IP instance, we use the GCN described in equations (3) to take as input an IP instance and output node embeddings, and we use the variable-node embeddings as label category embeddings respectively; (2) we use the state embedding $z_{s_t}^h$ of each LNS step as an input feature whose inner products with label embeddings correspond to feature-label similarity and can be used for prediction; (3) we use contrastive learning to capture correlations between label categories by pulling similar categories' embeddings together. Specifically, let $I \equiv \{1, \cdots, n\}$ and we define the positive label set of $a_t$ as $P(a_t) \equiv \{i \in I | a_t^i = 1\}$. Given a sequence of return-state-action tokens $(\langle g_{t-K}, s_{t-K}, a_{t-K}\rangle, \cdots, \langle g_t, s_t, a_t\rangle)$, our decoder is designed to optimize the following contrastive loss function:

$$\mathcal{L}_{CL} = \frac{1}{K} \sum_{j=t-K}^t \frac{1}{|P(a_j)|} \sum_{i \in P(a_j)} -\log \frac{z_{s_j}^h \cdot z_{a^i}^l}{\sum_{i' \in I} z_{s_j}^h \cdot z_{a^{i'}}^l}, \tag{5}$$

where state embeddings $z_{s_j}^h$ for $j \in [t-K, t]$ are generated by the causal transformer and are computed as in eq. (4).

For example, if in most of the actions $a_t$, labels $a_t^i$ and $a_t^j$ often appear together (i.e., they both equal 1), contrastive learning will implicitly pull their embeddings together. In other words, if two labels do co-appear often, their label embeddings would become similar. On the other hand, if they never co-occur or only co-appear occasionally, their connections are not significant and our decoder will not optimize for their similarity.

## 4.2 TRAINING ALGORITHM

Our model will be trained with supervised learning. Given a set of training IP instances, we first collect a dataset of sequences of return-state-action tokens that are generated by the MDP with some expert policy: $\mathcal{D} = \{(\langle g_{t-K}, s_{t-K}, a_{t-K}\rangle_j, \cdots, \langle g_t, s_t, a_t\rangle_j)\}_{j=1}^N$, where $K \leq T$ is the length of each sequence. For each sequence $q \in \mathcal{D}$, our model will take as input $q$ and generate a set of action predictions $\{\hat{a}_j\}_{j=t-K}^t$, and we will also collect the set of labels from $q$, $\{a_j\}_{j=t-K}^t$. A supervised cross-entropy loss for each sequence is given by

$$\mathcal{L}_{CE} = \frac{1}{K} \sum_{j=t-K}^t \frac{1}{n} \sum_{i=1}^n a_j^i \log \hat{a}_j^i + (1 - a_j^i) \log(1 - \hat{a}_j^i). \tag{6}$$

The final objective function to minimize is given by

$$\mathcal{L} = \mathcal{L}_{CL} - \beta \mathcal{L}_{CE}, \tag{7}$$

where $\beta$ is a trade-off weight. The model is trained with Adam Kingma and Ba [2014] and optimized with $\mathcal{L}$. We

sample mini-batches of sequence length $K$ from the dataset $\mathcal{D}$ and the model is trained with GPUs in parallel. However, similar to DT, our model will generate action predictions autoregressively during testing. We refer the reader to section 3.3 for more details.

# 5 EXPERIMENTS

We conduct experiments on four diverse NP-hard COP benchmarks, including minimum vertex cover (MVC), maximum cut (MC), set covering (SC), and combinatorial auction (CA). We follow the experimental settings of Song et al. [2020] and Wu et al. [2021].

## 5.1 DATASETS AND EXPERIMENTAL SETUP

**Datasets.** MVC and MC are graph optimization problems; SC and CA are general IPs. For MVC, we use the Erdős-Rényi (ER) model Erdős et al. [1960] to generate random graphs with 1000 nodes and edge probability 0.15. For MC, we use the Barabasi-Albert (BA) model Albert and Barabási [2002] to generate random graphs with 500 nodes and an average degree of 4. For SC, we generate instances with matrices having 5000 rows and 1000 columns following the procedure in Balas and Ho [1980], where each entry $B_{ij} \in \{0, 1\}$ represents whether the $i$-th element in the universe belongs to the $j$-th set. For CA, we use the Combinatorial Auction Test Suit (CATS) Leyton-Brown et al. [2000] with arbitrary relationships to generate instances with 2000 items and 4000 bids. For each problem type, we generate 100, 20, and 50 instances for training, validation, and testing, respectively. For each training instance, we use LNS with LB to run on it and set a time limit of half an hour for solving the sub-IP in each step with LB and Gurobi. We use the resulting trajectory for training ILP-FORMER, and we randomly sample 20 sequences of return-state-action tokens from each trajectory. Therefore, our dataset for training ILP-FORMER includes 2000 sequences of three tokens.

**Initialization.** LNS starts with a feasible initial solution. For MVC, MAXCUT, SC, and CATS, we initialize a feasible solution by including all vertices in the cover set, randomly partitioning all vertices into two complementary sets, including all sets in the set cover, and accepting no bids, respectively. The initialization process does not incur additional computational costs in our experiments.

**Implementation and Hyperparameters.** Return, action, and position encoders are all simple MLPs with 2 layers and 128 hidden neurons. The state encoder is a GCN with 2 convolution layers and a mean pooling layer. We use the GPT2 as our casual transformer. Dimensions of all hidden embeddings are set to 128. We set the batch size and the number of training epochs to 128 and 100, respectively, for all experiments. Our model was implemented with the

Pytorch deep learning framework and the whole model was trained using the Adam optimizer Kingma and Ba [2014] with a learning rate of 0.0001 and a weight decay ratio of 0.01 in an end-to-end fashion. All experiments were carried out on a machine with a 4.2 GHz quad-core Intel i7 CPU, 16 GB RAM, and an Nvidia RTX 3090 24GB GPU card.

The hyperparameters we employed are as follows: (1) Number of layers: 3; (2) Number of attention heads: 1; (3) Embedding dimension: 128; (4) Nonlinearity function: ReLU; (5) Batch size: 128; (6) Context length K: 25; (7) Dropout ratio: 0.1; (8) Learning rate: 1e-4; (9) Gradient norm clipping: 0.25. We maintained other parameters at their default values. We trained the model from scratch and did not utilize any pre-trained weights.

Grid search is adopted for tuning. We tune learning rate from 0.00005 to 0.002 with interval 0.00005, dropout ratio from [0.05, 0.1, 0.3, 0.5], weight decay from [0, 0.01, 0.0001], $\beta$ from [0.1, 0.5, 1, 1.5, 2.0], token embedding size from [64, 128, 256, 512], context length $K$ from [15, 20, 25, 30], batch size from [64, 128, 256], Gradient norm clipping from [0.15, 0.2, 0.25, 0.3, 0.35].

In the data collection process, we run LNS with LB and adaptive neighborhood size. The neighborhood size is initially set to 10% of the number of variables in the input problem instance. It is then adapted following the approach described in the paper by Sonnerat et al. [2021]. During testing, at each step $t$, the model generates action distributions $a_{t,i}$ for each dimension $i$ autoregressively. We apply a threshold of 0.5 to convert these values into 1 or 0, representing the selection or non-selection of the corresponding variable $x_i$ in step $t$.

**Baselines.** We compare our method with five baselines: (1) Gurobi (version 9.5) with default settings: a leading state-of-the-art IP solver; (2) FT-LNS: the best-performing LNS version by Song et al. [2020], which applies imitation learning to mimic the best demonstrations; (3) RL-LNS: the current state-of-the-art learning-based LNS method for solving ILPs Wu et al. [2021], which uses deep RL to learn LNS policy via action factorization to represent all potential variable subsets; (4) LB-SRMRL: the best-performing LB version by Liu et al. [2022], which uses a regression model and RL to learn a hybrid model to predict and adapt the neighborhood size for the LB heuristic; and (5) CL-LNS: the current state-of-the-art learning-based LNS method for solving ILPs Huang et al. [2023], which contrastive learning for ILP representation learning. We follow the default settings of these learning-based baselines and further fine-tune them on our datasets to get the best hyperparameters. For more details of the settings of these baselines, we refer the reader to their original papers.

**Evaluation Metrics.** The performances of different algorithms are compared in two measures: (1) the objective of solutions returned by different algorithms within a time

| Methods | MVC-1000 | | MC-500 | | SC-1000 | | CA-2000 | |
|---|---|---|---|---|---|---|---|---|
| | Obj.±Std.% | Gap% | Obj.±Std.% | Gap% | Obj.±Std.% | Gap% | Obj.±Std.% | Gap% |
| Gurobi | $482.2 \pm 0.8$ | 10.83 | $-863.9 \pm 3.8$ | 4.94 | $554.9 \pm 8.3$ | 6.26 | $-111668 \pm 2.0$ | 4.18 |
| FT-LNS | $470.0 \pm 0.4$ | 8.02 | $-866.2 \pm 1.7$ | 4.69 | $564.1 \pm 8.4$ | 8.02 | $-110041 \pm 1.6$ | 5.57 |
| RL-LNS | $469.0 \pm 0.5$ | 7.79 | $-878.0 \pm 1.6$ | 3.39 | $551.9 \pm 8.3$ | 5.69 | $-111787 \pm 2.6$ | 4.07 |
| LB-SRMRL | $472.4 \pm 0.7$ | 8.57 | $-859.1 \pm 2.3$ | 5.47 | $560.9 \pm 7.3$ | 7.41 | $-110741 \pm 3.1$ | 4.97 |
| CL-LNS | $450.2 \pm 0.4$ | 3.47 | $-865.3 \pm 1.6$ | 4.46 | $540.2 \pm 7.4$ | 3.45 | $-112956 \pm 2.1$ | 3.07 |
| ILP-FORMER | $\mathbf{435.1 \pm 0.8}$ | **0** | $\mathbf{-908.8 \pm 1.3}$ | **0** | $\mathbf{522.2 \pm 5.3}$ | 0 | $\mathbf{-116535 \pm 2.1}$ | **0** |

Table 1: A comparison of ILP-FORMER and the state-of-the-art baselines on 4 diverse benchmarks. The time limit is set to 200s. Each result is averaged over 5 runs. The gap is the ratio of objective difference w.r.t. the best result.

| Methods | MVC-2000 | | MC-1000 | | SC-2000 | | CA-4000 | |
|---|---|---|---|---|---|---|---|---|
| | Obj.±Std.% | Gap% | Obj.±Std.% | Gap% | Obj.±Std.% | Gap% | Obj.±Std.% | Gap% |
| Gurobi | $392.5 \pm 1.3$ | 10.53 | $-1784.7 \pm 1.0$ | 6.74 | $295.7 \pm 7.9$ | 6.63 | $-212890 \pm 1.8$ | 8.97 |
| FT-LNS | $390.5 \pm 1.1$ | 9.97 | $-1767.8 \pm 1.0$ | 7.62 | $303.3 \pm 8.0$ | 9.38 | $-211324 \pm 2.1$ | 9.64 |
| RL-LNS | $375.8 \pm 2.1$ | 5.83 | $-1831.0 \pm 0.9$ | 4.32 | $295.4 \pm 7.8$ | 6.53 | $-216650 \pm 1.7$ | 7.36 |
| LB-SRMRL | $395.2 \pm 1.9$ | 11.29 | $-1765.6 \pm 1.5$ | 7.73 | $301.4 \pm 7.2$ | 8.69 | $-209420 \pm 2.1$ | 10.45 |
| CL-LNS | $370.2 \pm 1.4$ | 4.25 | $-1865.3 \pm 1.6$ | 2.52 | $290.2 \pm 7.4$ | 4.65 | $-222956 \pm 2.1$ | 4.67 |
| ILP-FORMER | $\mathbf{355.1 \pm 1.1}$ | **0** | $\mathbf{-1913.6 \pm 0.8}$ | **0** | $\mathbf{277.3 \pm 7.1}$ | **0** | $\mathbf{-233870 \pm 1.7}$ | **0** |

Table 2: Generalization to larger instances with a double number of variables. The time limit is set to 500s.

| Methods | MVC-4000 | | MC-2000 | | SC-4000 | | CA-8000 | |
|---|---|---|---|---|---|---|---|---|
| | Obj.±Std.% | Gap% | Obj.±Std.% | Gap% | Obj.±Std.% | Gap% | Obj.±Std.% | Gap% |
| Gurobi | $278.3 \pm 0.9$ | 7.78 | $-3574.4 \pm 0.8$ | 5.91 | $175.4 \pm 7.0$ | 7.48 | $-422291 \pm 1.2$ | 4.71 |
| FT-LNS | $279.2 \pm 1.7$ | 8.13 | $-3526.2 \pm 0.8$ | 7.18 | $175.2 \pm 6.6$ | 7.35 | $-431234 \pm 0.9$ | 2.69 |
| RL-LNS | $273.6 \pm 2.1$ | 5.96 | $-3612.5 \pm 0.7$ | 4.91 | $172.4 \pm 7.1$ | 5.64 | $-432980 \pm 0.7$ | 2.30 |
| LB-SRMRL | $275.6 \pm 2.2$ | 6.74 | $-3505.1 \pm 0.9$ | 7.73 | $177.1 \pm 7.2$ | 8.52 | $-415631 \pm 0.5$ | 6.21 |
| CL-LNS | $270.2 \pm 2.4$ | 4.65 | $-3535.3 \pm 0.6$ | 6.94 | $173.2 \pm 7.1$ | 4.92 | $-434211 \pm 2.1$ | 2.02 |
| ILP-FORMER | $\mathbf{258.2 \pm 1.9}$ | **0** | $\mathbf{-3798.9 \pm 1.0}$ | **0** | $\mathbf{163.2 \pm 6.1}$ | 0.60 | $\mathbf{-439151 \pm 0.5}$ | **0** |

Table 3: Generalization to larger instances with a quadruple number of variables. The time limit is set to 500s.

limit; (2) the gap between solutions, namely, the ratio of objective difference w.r.t. the best result.

## 5.2 EXPERIMENTAL RESULTS

A comparison of ILP-FORMER and other state-of-the-art baselines on 4 diverse benchmarks is given in Table 1. All learning-based algorithms, including our ILP-FORMER, call Gurobi to solve sub-IPs with a time limit of 2s at every step. We can observe that LB-SRMRL is not comparable to other algorithms. CL-LNS remains the most competitive baseline and consistently outperforms Gurobi, RL-LNS, and FT-LNS. Overall, these results suggest that our approach can reliably offer substantial improvements over state-of-the-art solvers.

We also compare the generalization ability of all algorithms to solve large IPs. To this end, we generate two sets of testing instances following the same settings as in section 5.1 but double and quadruple the number of variables respectively. Note that we only generate 50 testing instances for each problem type without considering training and validation. We test all (trained) models on these new instances and summarize results in Tables 6 and 3. We can observe that the advantage of our ILP-FORMER still preserves on larger problem instances compared to baselines. Specifically, Table 6 shows that ILP-FORMER still consistently outperforms all baselines on the 4 benchmarks when the in-

stance size is doubled. On the other hand, Table 3 shows that ILP-FORMER outperforms all baselines on 3 out of 4 benchmarks when the instance size is quadrupled. In summary, our ILP-FORMER learned on small instances generalizes well to larger instances, with a persistent advantage over other methods.

## 5.3 ANYTIME PERFORMANCE

We further showcase the anytime performance of various algorithms, including random LNS in this experiment, to facilitate an easier comparison between random LNS and ILP-FORMER across four benchmarks, as illustrated in Figure 3. Our observations indicate that: (1) ILP-FORMER significantly outperforms other baselines with a noteworthy margin. (2) Even with extended time limits, ILP-FORMER's advantage persists.

## 5.4 ADDITIONAL EXPERIMENTS WITH SCIP

Our framework can integrate any ILP solver to enhance incumbent solutions. We primarily conducted experiments with Gurobi, given its status as a leading ILP solver. Additionally, we also present results utilizing SCIP (v6.0.1) as an alternative ILP solver. By employing the same settings as detailed in Section 5.1 and applying them to the four benchmarks, we display the results in Table 5. These

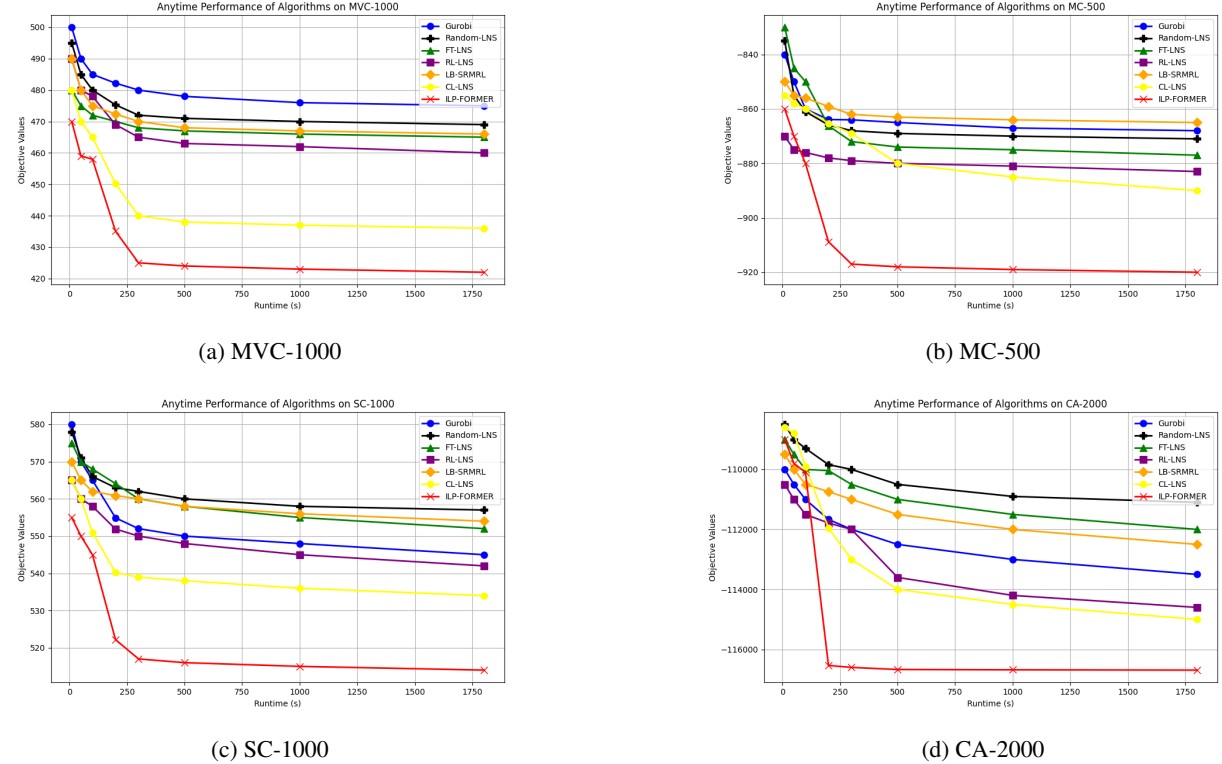

Figure 3: Anytime Performance Comparison of Gurobi, FT-LNS, RL-LNS, LB-SRMRL, CL-LNS, and ILP-FORMER on Four IP benchmarks. Runtimes are up to 30 minutes.

| Methods | MVC-1000 | | MC-500 | | SC-1000 | | CA-2000 | |
|---------|----------|------|--------|------|---------|------|---------|------|
| | Obj.±Std.% | Gap% | Obj.±Std.% | Gap% | Obj.±Std.% | Gap% | Obj.±Std.% | Gap% |
| SCIP | $552.2 \pm 0.5$ | 21.60 | $-793.9 \pm 2.8$ | 11.41 | $604.3 \pm 3.2$ | 11.41 | $-100514 \pm 2.1$ | 12.31 |
| FT-LNS | $490.2 \pm 0.6$ | 7.95 | $-836.3 \pm 1.2$ | 6.68 | $585.2 \pm 6.4$ | 7.91 | $-107141 \pm 1.7$ | 6.53 |
| RL-LNS | $480.0 \pm 0.5$ | 5.70 | $-847.5 \pm 1.3$ | 2.42 | $575.6 \pm 6.2$ | 6.14 | $-108787 \pm 2.2$ | 5.09 |
| LB-SRMRL | $492.4 \pm 0.9$ | 8.43 | $-820.3 \pm 1.9$ | 8.47 | $580.8 \pm 5.3$ | 7.10 | $-107741 \pm 3.0$ | 6.01 |
| CL-LNS | $470.4 \pm 0.9$ | 3.66 | $-860.3 \pm 1.9$ | 4.01 | $560.8 \pm 5.3$ | 3.41 | $-109941 \pm 3.0$ | 4.09 |
| ILP-FORMER | $\mathbf{454.1 \pm 0.7}$ | **0** | $\mathbf{-896.2 \pm 1.2}$ | **0** | $\mathbf{542.3 \pm 5.3}$ | 0 | $\mathbf{-114626 \pm 1.5}$ | **0** |

Table 4: Results with SCIP. The time limit is set to 200s. Each result is averaged over 5 runs. The gap is the ratio of objective difference w.r.t. the best result. The best results are shown in **bold**.

outcomes align with those observed when using Gurobi as the ILP solver, albeit with SCIP exhibiting a notably lower performance compared to Gurobi.

## 5.5 ABLATION STUDY

To demonstrate the strength of ILP-FORMER, we compare it with two variants: (1) ILP-FORMER⊖MLC: a modified ILP-FORMER where its MLC decoder is replaced with a linear decoder; (2)ILP-FORMER⊖DT: a modified ILP-FORMER where its casual transformer component is removed. The results are summarized in Table 5. We can observe that ILP-FORMER outperforms the two variants consistently; our model's performance drop significantly if we do not consider modeling the sequential process of LNS (drop by 4.09% on average) or exploit correlations of variable selection (drop by 3.13% on average).

## 5.6 TESTING ON REAL-WORLD INSTANCES IN MIPLIB

We follow the experimental settings for real-world instances in MIPLIB as described in Wu et al. [2021]. We exclude "easy" instances with relatively small sizes, as well as instances where Gurobi cannot find any feasible solutions within a 3600-second time limit. Consequently, we choose 35 representative "hard" or "open" instances containing only integer variables. Within these instances, the number of variables ranges from 150 to 393,800 (averaging 49,563), and the number of constraints varies from 301 to 850,513 (averaging 96,778). We use the datasets in section 5.1 to train our model and evaluate our model (with Gurobi as the repair solver) on this realistic dataset, in the style of *active search* Bello et al. [2016], Wu et al. [2021], Khalil et al. [2017] on each instance. Our findings indicate that, with a 3600-second time limit, ILP-FORMER surpasses both

| Methods | MVC-1000 | | MC-500 | | SC-1000 | |
|---|---|---|---|---|---|---|
| | Obj.±Std.% | Gap% | Obj.±Std.% | Gap% | Obj.±Std.% | Gap% |
| ILP-FORMER⊖MLC | $460.5 \pm 0.6$ | 5.86 | $-882.0 \pm 1.5$ | 2.95 | $540.2 \pm 8.3$ | 3.45 |
| ILP-FORMER⊖DT | $452.6 \pm 1.1$ | 4.02 | $-889.1 \pm 0.8$ | 2.17 | $538.9 \pm 6.3$ | 3.20 |
| ILP-FORMER | $\mathbf{435.1 \pm 0.8}$ | **0** | $\mathbf{-908.8 \pm 1.3}$ | **0** | $\mathbf{522.2 \pm 5.3}$ | 0 |

Table 5: An ablation study on the casual transformer and MLC decoder components of ILP-FORMER. Note that the experimental settings here follow that of Table 1.

solvers on 20 of the 35 instances and exhibits comparable performance on 10 of the 35 instances.

# 6 CONCLUSION

This paper concentrates on enhancing learning-based LNS approaches, given their ability to conveniently utilize any existing solver as a subroutine. Thus, they can inherit the advantages of meticulously engineered heuristic or complete approaches, along with their software implementations. We introduce ILP-FORMER, a novel approach that models policy learning as a sequence to an MLC problem. It seamlessly integrates a customized decision transformer encoder, encompassing a causal transformer, to model the sequential processes of LNS, and an MLC decoder with contrastive learning to exploit correlations in variable selection. Furthermore, we carry out comprehensive experiments on diverse benchmarks. The results suggest that our ILP-FORMER approach consistently delivers substantial improvements over state-of-the-art solvers and exhibits excellent generalization capabilities for larger instances.

## ACKNOWLEDGMENT

This project is partially supported by the Eric and Wendy Schmidt AI in Science Postdoctoral Fellowship, a Schmidt Futures program; the National Science Foundation, the Air Force Offce of Scientifc Research; the Department of Energy (DOE); and the Toyota Research Institute (TRI). The work of Caihua Liu is supported and funded by the Humanities and Social Sciences Youth Foundation, Ministry of Education of the People's Republic of China (Grant No.21YJC870009).

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

| Instance | Gurobi | CL-LNS | ILP-FORMER |
|---|---|---|---|
| a2864-99blp | -72 | -73 | **-85** |
| bab3 | -655388.1120 | -655022.5305 | **-655412.3501** |
| bley-xs1noM | **3938322.37** | 3965411.35 | 3958310.21 |
| cdc7-4-3-2 | -257 | -276 | **-280** |
| comp12-2idx | 380 | 363 | **352** |
| ds | **177** | 319 | 189 |
| ex1010-pi | 239 | 238 | 238 |
| graph20-80-1rand | -6 | -6 | -6 |
| graph40-20-1rand | **-15** | -14 | -14 |
| neos-3426085-ticino | 226 | 226 | 226 |
| neos-3594536-henty | 401948 | 402426 | **401896** |
| neos-3682128-sandon | 34666770.0 | **34666765.12338** | 34666770 |
| ns1828997 | 133 | 128 | **98** |
| nursesched-medium-hint03 | 115 | 131 | 115 |
| opm2-z12-s8 | -33269 | -53379 | **-55269** |
| pb-grow22 | -46217.0 | -46881.0 | **-56782** |
| proteindesign121hz512p9 | 1499 | 1489 | **1481** |
| queens-30 | -39 | -39 | -39 |
| ramos3 | 245 | 248 | **216** |
| rmine13 | -3493.781904 | -3487.807859 | **-3493.79821** |
| rmine15 | -1979.559046 | -5001.279118 | **-5002.129874** |
| rococoC12-010001 | 34673 | **35440** | 35467 |
| s1234 | 29 | 40 | 29 |
| scpj4scip | 133 | 134 | **131** |
| scpk4 | 330 | 329 | **325** |
| scpl4 | 279 | 281 | **269** |
| sorrell3 | -16 | -16 | -16 |
| sorrell4 | -23 | -23 | -23 |
| sorrell7 | -187 | -187 | **-190** |
| supportcase2 | 397 | 397 | 397 |
| t1717 | 201342 | 186891 | **185241** |
| t1722 | 117171 | 117978 | **115983** |
| tokyometro | 8479.5 | 8562.80 | **8456.7** |
| v150d30-2hopcds | 41 | 41 | 41 |
| z26 | -1083 | -1172 | **-1176** |

Table 6: Results on MIPLIB. The best results are shown in **bold**. The time limit is set to 3600s.

Yanchen Deng, Shufeng Kong, Caihua Liu, and Bo An. Deep attentive belief propagation: Integrating reasoning and learning for solving constraint optimization problems. *arXiv preprint arXiv:2209.12000*, 2022b.

Dorian Dumez, Fabien Lehuédé, and Olivier Péton. A large neighborhood search approach to the vehicle routing problem with delivery options. *Transportation Research Part B: Methodological*, 144:103–132, 2021.

Paul Erdős, Alfréd Rényi, et al. On the evolution of random graphs. *Publ. Math. Inst. Hung. Acad. Sci*, 5(1):17–60, 1960.

Maxime Gasse, Didier Chételat, Nicola Ferroni, Laurent Charlin, and Andrea Lodi. Exact combinatorial optimization with graph convolutional neural networks. *Advances in Neural Information Processing Systems*, 32, 2019.

Stefan Gollowitzer and Ivana Ljubić. Mip models for connected facility location: A theoretical and computational study. *Computers & Operations Research*, 38(2):435–449, 2011.

Taoan Huang, Aaron Ferber, Yuandong Tian, Bistra Dilkina, and Benoit Steiner. Searching large neighborhoods for integer linear programs with contrastive learning. In *Proceedings of the 40th International Conference on Machine Learning*, ICML'23. JMLR.org, 2023.

Chaitanya K Joshi, Thomas Laurent, and Xavier Bresson. An efficient graph convolutional network tech-

nique for the travelling salesman problem. *arXiv preprint arXiv:1906.01227*, 2019.

Leslie Pack Kaelbling, Michael L Littman, and Andrew W Moore. Reinforcement learning: A survey. *Journal of artificial intelligence research*, 4:237–285, 1996.

Richard M Karp. Reducibility among combinatorial problems. In *Complexity of computer computations*, pages 85–103. Springer, 1972.

Elias Khalil, Hanjun Dai, Yuyu Zhang, Bistra Dilkina, and Le Song. Learning combinatorial optimization algorithms over graphs. *Advances in neural information processing systems*, 30, 2017.

Diederik P Kingma and Jimmy Ba. Adam: A method for stochastic optimization. *arXiv preprint arXiv:1412.6980*, 2014.

Wen-Yang Ku and J Christopher Beck. Mixed integer programming models for job shop scheduling: A computational analysis. *Computers & Operations Research*, 73: 165–173, 2016.

Eric Larsen, Sébastien Lachapelle, Yoshua Bengio, Emma Frejinger, Simon Lacoste-Julien, and Andrea Lodi. Predicting solution summaries to integer linear programs under imperfect information with machine learning. *arXiv preprint arXiv:1807.11876*, 2018.

Kevin Leyton-Brown, Mark Pearson, and Yoav Shoham. Towards a universal test suite for combinatorial auction algorithms. In *Proceedings of the 2nd ACM conference on Electronic commerce*, pages 66–76, 2000.

Defeng Liu, Matteo Fischetti, and Andrea Lodi. Learning to search in local branching. In *Proceedings of the AAAI Conference on Artificial Intelligence*, volume 36, pages 3796–3803, 2022.

Josefa Mula, Raul Poler, Jose P García-Sabater, and Francisco Cruz Lario. Models for production planning under uncertainty: A review. *International journal of production economics*, 103(1):271–285, 2006.

Vinod Nair, Mohammad Alizadeh, et al. Neural large neighborhood search. In *Learning Meets Combinatorial Algorithms at NeurIPS2020*, 2020a.

Vinod Nair, Sergey Bartunov, Felix Gimeno, Ingrid von Glehn, Pawel Lichocki, Ivan Lobov, Brendan O'Donoghue, Nicolas Sonnerat, Christian Tjandraatmadja, Pengming Wang, et al. Solving mixed integer programs using neural networks. *arXiv preprint arXiv:2012.13349*, 2020b.

Laurent Perron, Paul Shaw, and Vincent Furnon. Propagation guided large neighborhood search. In *International Conference on Principles and Practice of Constraint Programming*, pages 468–481. Springer, 2004.

David Pisinger and Stefan Ropke. Large neighborhood search. In *Handbook of metaheuristics*, pages 399–419. Springer, 2010.

Alec Radford, Jeffrey Wu, Rewon Child, David Luan, Dario Amodei, Ilya Sutskever, et al. Language models are unsupervised multitask learners. *OpenAI blog*, 1(8):9, 2019.

Yasaman Razeghi, Kalev Kask, Yadong Lu, Pierre Baldi, Sakshi Agarwal, and Rina Dechter. Deep bucket elimination. In *IJCAI*, pages 4235–4242, 2021.

Stefan Ropke and David Pisinger. An adaptive large neighborhood search heuristic for the pickup and delivery problem with time windows. *Transportation science*, 40(4): 455–472, 2006.

Stéphane Ross and Drew Bagnell. Efficient reductions for imitation learning. In *Proceedings of the thirteenth international conference on artificial intelligence and statistics*, pages 661–668. JMLR Workshop and Conference Proceedings, 2010.

Jialin Song, Yisong Yue, Bistra Dilkina, et al. A general large neighborhood search framework for solving integer linear programs. *Advances in Neural Information Processing Systems*, 33:20012–20023, 2020.

Nicolas Sonnerat, Pengming Wang, Ira Ktena, Sergey Bartunov, and Vinod Nair. Learning a large neighborhood search algorithm for mixed integer programs. *arXiv preprint arXiv:2107.10201*, 2021.

Richard S Sutton and Andrew G Barto. *Reinforcement learning: An introduction*. MIT press, 2018.

Hamdy A Taha. *Integer programming: theory, applications, and computations*. Academic Press, 2014.

Faraz Torabi, Garrett Warnell, and Peter Stone. Behavioral cloning from observation. In *Proceedings of the 27th International Joint Conference on Artificial Intelligence*, pages 4950–4957, 2018.

Ashish Vaswani, Noam Shazeer, Niki Parmar, Jakob Uszkoreit, Llion Jones, Aidan N Gomez, Łukasz Kaiser, and Illia Polosukhin. Attention is all you need. *Advances in neural information processing systems*, 30, 2017.

Ronald J Williams. Simple statistical gradient-following algorithms for connectionist reinforcement learning. *Machine learning*, 8(3):229–256, 1992.

Yaoxin Wu, Wen Song, Zhiguang Cao, and Jie Zhang. Learning large neighborhood search policy for integer programming. *Advances in Neural Information Processing Systems*, 34:30075–30087, 2021.

Si Zhang, Hanghang Tong, Jiejun Xu, and Ross Maciejewski. Graph convolutional networks: a comprehensive review. *Computational Social Networks*, 6(1):1–23, 2019.
