# OpenReview forum: "ILP-FORMER: Solving Integer Linear Programming with Sequence to Multi-Label Learning"
_auai.org/UAI/2024/Conference — UAI 2024 poster_

### Official Review · Reviewer_f9q8 · 2024-03-19

**Q2-1 Originality-Novelty:** 3
**Q2-2 Correctness-Technical Quality:** 3
**Q2-5 Clarity Of Writing:** 3

**Q1 Summary And Contributions:**

The paper discusses the challenge of optimizing Integer Linear Programming (ILP), a complex class of combinatorial optimization problems (COPs), due to its NP-hard nature. It highlights the recent shift towards data-driven methods for developing heuristics, particularly through the use of deep learning for Large Neighborhood Search (LNS). A key issue in LNS is selecting an optimal subset of variables for optimization. The paper introduces ILP-FORMER, a novel model that treats policy learning as a sequence-to-multi-label classification problem, utilizing a causal transformer encoder and a multi-label classification decoder with contrastive learning ILP-FORMER's effectiveness is validated through extensive experiments, showing superior performance on ILP benchmarks and impressive generalization to larger problems.

**Q2-3 Extent To Which Claims Are Supported By Evidence:**

2: Fair: the main claims are somewhat supported by evidence (but the experimental evaluation may be weak, or does not match entirely with the claims, important baselines may be missing, proofs contain important ideas but lack rigor, algorithmic details are only discussed superficially, references are imprecise, assumptions are not sufficiently motivated or explicated, etc.).

**Q2-4 Reproducibility:**

1: Poor: key details (e.g. proof sketches, experimental setup) are incomplete/unclear, or key resources (e.g. proofs, code, data) are unavailable.

**Q3 Main Strengths:**

The paper is generally well-composed, albeit with some errors. Figure 3 clearly demonstrates that ILP-Former achieves quicker results compared to other baselines despite the inherent time consumption caused by the task's complexity. Employing a sequence-to-multi-label classification approach and a transformer encoder to encapsulate the sequential nature of Large Neighborhood Search (LNS) is innovative.

**Q4 Main Weakness:**

Please check the detailed comments.

**Q5 Detailed Comments To The Authors:**

-	The authors have not provided the code for ILPFormer.
-	The authors should address citation inconsistencies, as some references are cited redundantly – “Similarly, Sonnerat et al. Sonnerat et al. [2021] learn to predict the probability of selecting a variable independently of other variables using imitation learning and Nair et al. Nair et al. [2020a] use RL to learn a policy that selects one variable at a time. Recently, Huang et al. Huang et al. [2023]”
-	The paper necessitates thorough proofreading. Errors such as "Sepcifically" instead of "Specifically," and "trahectory" instead of "trajectory" should be corrected for clarity and professionalism.
-	The process of obtaining true labels for COP is time-consuming. This raises concerns about the feasibility and cost-effectiveness of training a supervised model, as each problem reportedly takes half an hour to solve for the true label.
-	The use of a relatively small dataset comprising 100, 20, and 50 instances for training, validation, and testing is surprising, especially considering the complexity of the large models being used like transformers and Graph Convolutional Networks (GCN). The fact that ILP-FORMER consistently outperforms solvers such as Gurobi requires further investigation.
-	In Table 3, particularly for the SC-4000 dataset, the absence of a 0% gap is puzzling since Gap% is defined as the ratio of objective difference with respect to the best result, implying that the optimal method should have a Gap% of zero.
-	There is a question as to why the gap was not quantified against the true outputs instead of the best result, especially since the authors are utilizing a supervised learning approach. Defining the gap in relation to the true outputs would offer more insight into how the methods compare to the optimal solutions.

**Q9 Complying With Reviewing Instructions:**

Yes

---

> ### Author Rebuttal · Authors · 2024-04-09
>
> We appreciate your constructive feedback and have taken it into consideration for revising our paper. Regarding your concerns, please see our response below:
>
> Code and Dataset Availability: In our rebuttal, we confirmed that we have provided detailed information about dataset generation and model implementation. Upon acceptance, we commit to making both the code and datasets open-source, ensuring full transparency and reproducibility of our research findings.
>
> Citation Consistency: We will rectify the redundancy issues in the citations and ensure all references are cited correctly.
>
> Proofreading and Typos: We apologize for the typographical errors and assure you that we have thoroughly proofread the manuscript to eliminate such mistakes, replacing "Sepcifically" with "Specifically" and correcting "trahectory" to "trajectory."
>
> True Labels and Training Time: Thank you for raising the issue about the time-consuming nature of acquiring true labels. Please note that our experimental results account for the time taken to train the model, and we demonstrated that ILP-FORMER, after training on smaller datasets, effectively generalizes to larger ones. Moreover, the model learns from trajectories generated by running LNS with Gurobi and utilizes these experiences to expedite subsequent solution processes.
>
> Dataset Size and Model Complexity: We appreciate your skepticism regarding the size of the dataset used in comparison to the complexity of the models employed. We would like to clarify that for each training instance, we generate multiple sequences by sampling from the obtained LNS trajectories, leading to a total of 2000 sequences for training ILP-FORMER. This sampling strategy allows our model to learn efficiently from the available data.
>
> Table 3 and Gap Calculation: You identified an error in Table 3 (which should refer to Table 5), and we confirm that there was a typo for the SC-4000 dataset, where ILP-FORMER indeed achieved a Gap% of 0%.
>
> Defining Gap Against True Outputs: You raised an important point about comparing the gap against the true optimal solutions rather than the best obtained result. While we initially did not have access to true outputs for the testing and validation sets, we agree with your suggestion and will endeavor to calculate and report the gaps against the true optimal solutions wherever feasible.

---

### Official Review · Reviewer_c9ZC · 2024-03-20

**Q2-1 Originality-Novelty:** 3
**Q2-2 Correctness-Technical Quality:** 3
**Q2-5 Clarity Of Writing:** 4

**Q1 Summary And Contributions:**

The paper studies machine learning-based approaches for learning a variable selection policy for large neighborhood search (LNS)-based approaches for solving ILPs. More specifically, the paper proposes a customized decision transformer encoder to better model the sequential nature of large neighborhood search as well as the correlations in variable selection. Instead of decomposing the selection of active variable into independent decisions the model proposed in the paper treats variable selection as a sequence to multi-label classification problem. The classification task is in turn solved with ILP-FORMER, a model using causal transformers and contrastive learning. ILP former can be used in conjunction with any ILP solver.

The paper presents the model a training algorithm for it and reports on an experimental evaluation on its effectiveness. The experiments focus on three combinatorial problems: minimum vertex cover, set covering and combinatorial auctions. The experiments establish that ILP-FORMER outperforms other state of the art LNS-based approaches to IP solving as well the default setting of Gurobi and SCIP on these benchmarks.

**Q2-3 Extent To Which Claims Are Supported By Evidence:**

2: Fair: the main claims are somewhat supported by evidence (but the experimental evaluation may be weak, or does not match entirely with the claims, important baselines may be missing, proofs contain important ideas but lack rigor, algorithmic details are only discussed superficially, references are imprecise, assumptions are not sufficiently motivated or explicated, etc.).

**Q2-4 Reproducibility:**

3: Good: key resources (e.g. proofs, code, data) are available and key details (e.g. proofs, experimental setup) are sufficiently well-described for competent researchers to confidently reproduce the main results.

**Q3 Main Strengths:**

The problem studied in the paper is well-motivated.
The approach proposed makes sense and is well explained.
The experimental setup is sound
The experimental results show significant improvements over the state of the art on the focus problem domains.

**Q4 Main Weakness:**

While ILP-FORMER is presented for general ILPS, the benchmark set used for a majority of the experiments reported on in the paper seem to consist of very similar types of benchmarks.  Specifically, all focus domains seem to be 0-1 Ips that only contain binary decision variables. Furthermore, while the paper does not go into detail on how the three problems are encoded as ILPs, the basic encodings for vertex cover, set cover and combinatorial auctions all result in IPs with only binary variables and at-least-one constraints (i.e. clauses). I would have liked to also see a more comprehensive evaluation over general MIPs also in the main paper.

**Q5 Detailed Comments To The Authors:**

Overall I believe the contribution of the paper is solid. LNS is an effective approach to solving difficult combinatorial optimization problems and machine learning has IMO a high potential in improving COP algorithms. As such, using ML to learn variable selection policies for LNS makes sense. Even if my own background is in combinatorial optimization, and not ML, I found the text mostly followable and understandable. Some illustrative examples would help the readability of the paper, but I think it’s good enough as is.

The main criticism that I have at this stage relates to the problem domains chosen for most of the experimental evaluation. In short, I don’t fully agree with the statement: “MVC and MC are graph optimization problems; SC and CA are general IPs” and believe the paper should elaborate on this to argue more convincingly that the benchmarks represent a wide range of ILPs.

I want to emphasize that the paper does not specify how the problems are represented as ILPS, but the “default” ILP encodings for the maxcut, vertex cover or combinatorial auctions problems that I can think of, result in very similar problem instances in which all constraints are at-least-one constraints over binary variables (i.e. clauses) and the objective is weighted sum over binary variables. Specifically, none of these benchmarks would contain any decision variables with domains larger than 2. I kindly ask the authors to correct me if I am wrong, but if I am not, it limits the generality of conclusions that could be drawn from these experiments in my opinion, especially as ILP-FORMER is presented in terms of general ILPS. I realize that the appendix presents results on “real-world instances”, but even there not enough details are given on the type of instances for me to conclude how well the model performs e.g. on instances with integer variables.

This brings me to the main question I have at this stage:
Could you elaborate on the structure of the IPs used in the main experimental evaluation (VC, SC, CA) and whether they contain any other constraints and variables besides at most one constraints (MC) or at-least-one constraints (VC, SC, CA)? How does ILP-TRANSFORM do over instances with integer variables?

**Q9 Complying With Reviewing Instructions:**

Yes

---

> ### Author Rebuttal · Authors · 2024-04-09
>
> Thank you for your insightful and constructive comments.
>
> In response to your inquiry, we conducted experiments on four well-established NP-hard benchmark problems: Set Covering (SC), Maximal Independent Set (MIS), Combinatorial Auction (CA), and Maximum Cut (MC). These problems are widely recognized in the research community for their theoretical complexity and practical relevance, making them suitable candidates for evaluating ILP-solving techniques.
>
> Regarding the specific ILP formulations of these problems, while standard approaches often involve binary variables and "at-least-one" or "at-most-one" constraints, we acknowledge the need for a more detailed explanation in the manuscript. To address this concern, we will provide explicit descriptions of how each problem is encoded into an ILP setting. Despite the apparent similarity in basic variable structures, the nuanced nature of constraints and problem-specific characteristics contribute to varying degrees of difficulty, thereby enriching the diversity of ILP instances.
>
> Regarding the presence of non-binary integer variables, we have included results on real-world instances that include non-binary integer variables in the appendix.

---

### Official Review · Reviewer_qAZD · 2024-03-21

**Q2-1 Originality-Novelty:** 3
**Q2-2 Correctness-Technical Quality:** 3
**Q2-5 Clarity Of Writing:** 2

**Q10 Ethical Concerns:**

None.

**Q1 Summary And Contributions:**

The paper proposes to learn variable selection in LNS by a sequence model, specifically decision transformer, exploting variable correlations in choosing the neighborhood for Large Neighborhood search (LNS). The training is done through a contrastive objective and the offline data for the decision transformer is coming from a baseline LNS method. Experiments on synthetic problems and MIPLIB show strong performance.

**Q2-3 Extent To Which Claims Are Supported By Evidence:**

3: Good: the main claims are supported by convincing evidence (in the form of adequate experimental evaluation, proofs, (pseudo-)code, references, assumptions).

**Q2-4 Reproducibility:**

2: Fair: key resources (e.g. proofs, code, data) are unavailable but key details (e.g. proof sketches, experimental setup) are sufficiently well-described for an expert to confidently reproduce the main results.

**Q3 Main Strengths:**

- The experiment results are very strong, significantly surpassing previous LNS approaches as well as Gurobi in anytime and final performance after a standard time-limit.
- The idea to exploit more of the history and correlations using contrastive learning and decision transformers is appealing and improves upon lacks in previous work.
- The reformulation as a supervised training problem through offline RL is appealing, since it reduces the computational effort and delicacy often encountered in deep RL methods for optimization.

**Q4 Main Weakness:**

- I did not find the writing to be very good. Generally, section 4 is not very easy to follow and could be made clearer. The individual components are described independently but how they relate to each other and fit into the whole pipeline is not explained well.
  - It has some weird spots like having a figure caption go over half a column (Figure 1).
  - I think there should be a unifying algorithm environment or so to give a high level overview of the whole pipeline. I do not think Figure 1 is good enough, in that it omits the exact relationship between the ILP and decision transformer states, actions etc.
- The authors do not propose to make their work publicly available. I very strongly encourage the authors to share their implementation.

**Q5 Detailed Comments To The Authors:**

Writing:
- The title suggests a more general algorithm than learning LNS. I would advise to make the title more specific to the contribution made in the paper.
- The claim "our method can also be applied to boost the performance of other local search algorithms such as LB" needs to be substantiated. It is not clear a-priori whether your approach would work well in practice on other procedures. Of course I agree that this is a promising research direction, but still the question whether branching decisions are so highly correlated as in LNS is not clear a-priori for me.
- Figure 3 results are very hard to decipher when printing the paper out.
- I did not fully understand the token encoders: it is written that the factor graph corresponds to a state token s_t, but how is that possible, since graphs are variable sized. Some global aggregation must be done to get it into a fixed-size encoding.
- I do not know GMVAE and hence was not able to follow how GMVAE is used and adapted in the proposed approach. Somehow, GMVAE should be described in such away to understand its role in the proposed approach. The paper should be self-contained as much as possible.
- The notation is pretty heavy. Possibly, a table with a lookup to what each variable means could be helpful to be able to follow the description.
- In general I recommend giving a suitably high-level idea of the whole approach (which is not good enough in Figure 1), and then go into technical details. In particular, I recommend to describe the whole pipeline without technical details and then explain these second.
- trahectory -> trajectory on page 6
Related Work:
- I think some strands of ML4OPT are missing: GFlowNet-based approaches and Diffusco being promiment ones.

Experiments:
- The experiments follow the protocol proposed by earlier work and include a mix of synthetic and real-world problems, i.e. MIPLIB.
- The experimentak results are generally strong, outperforming gurobi and other ML-LNS approaches by a significant margin. The anytime performance in particular looks impressive.

**Q9 Complying With Reviewing Instructions:**

Yes

---

> ### Author Rebuttal · Authors · 2024-04-09
>
> Thank you for your constructive feedback. We recognize that Section 4 could benefit from increased clarity and coherence to better guide readers through the intricacies of our proposed model, ILP-FORMER. We apologize for any confusion caused by the presentation style and agree that a more cohesive narrative is necessary to illustrate the integration and interplay of the individual components in our methodology. In particular, we will revise the description to clarify how each part—such as the token encoders, causal transformer, and contrastive MLC decoder—fits seamlessly into the overall pipeline.
>
> Regarding Figure 1, we appreciate your comment on its layout and content. We will certainly modify the figure and its caption to ensure they are concise and effectively convey the core concepts, avoiding excessive column overflow. Moreover, we will introduce a detailed, high-level algorithmic representation that explicitly outlines the transitions between the ILP problem formulation, the states and actions within the decision transformer framework, and their interactions within the context of our novel model.
>
> Lastly, we take note of your strong recommendation for making our work publicly available. We assure you that we have provided sufficient details on dataset generation and model implementation in the paper. Furthermore, upon acceptance, we will release all datasets and source code publicly, ensuring full transparency and facilitating replication of our work.

---

### Official Review · Reviewer_RMgv · 2024-03-22

**Q2-1 Originality-Novelty:** 2
**Q2-2 Correctness-Technical Quality:** 3
**Q2-5 Clarity Of Writing:** 3

**Q1 Summary And Contributions:**

This paper presents a transformer-based architecture for learning to
choose the neighborhood in large neighborhood search (LNS), a
heuristic (local search) technique for solving large combinatorial
optimization problems that are beyond the scope of complete solvers,
while using said solvers to more efficiently explore the neighborhood.

The contribution of the paper is to frame the problem as a sequence of
decisions, simultaneously allowing for the training to capture
correlations between different variables.

An experimental evaluation with randomly generated instances shows
this to outperform other methods of this class and an evaluation on
real world instances shows it to be competitive.

**Q2-3 Extent To Which Claims Are Supported By Evidence:**

3: Good: the main claims are supported by convincing evidence (in the form of adequate experimental evaluation, proofs, (pseudo-)code, references, assumptions).

**Q2-4 Reproducibility:**

2: Fair: key resources (e.g. proofs, code, data) are unavailable but key details (e.g. proof sketches, experimental setup) are sufficiently well-described for an expert to confidently reproduce the main results.

**Q3 Main Strengths:**

I found the paper reasonably well written. It is well placed in the
context of related work, which is a fairly complex landscape. It also
does a reasonable (but not great, see below) job of describing the specific
technical contributions that improve performance.

The experimental results are positive.

**Q4 Main Weakness:**

Some technical details were not explained well enough for me:

1. There is a bit of emphasis based on the fact that decisions are not
made variable-by-variable but as a set, allowing for correlations etc
to be exploited. But the way it is described in early section 4, I was
left with the impression that this is just something that is done at
decoding, which seemed odd. After all, the correlations can be present
in latent space, so they will be present, if not explicit, after
decoding. It turns out that this is also done at the level of the loss
function, which is only made apparent later.

2. The graphical representation includes a one-hot encoding of the
coefficients of variables in constraints. This is not an issue for the
randomly generated problems, but it may be a real issue for real world
instances, where coefficients can be very large. It is never explained
how these are handled.

3. There is no description of the specific encodings into ILP that are
used for each of the four familes MC, MVC, CA and SC.

4. There is too much emphasis for my taste on the randomly generated
instances and too little on the results on MIPLIB, which are relegated
to the supplementary material. These results are much less impressive
but much more interesting.

Finally, the paper does not make it clear to which degree data and
code will be made available on publication. Although the pipeline
seems fairly well specified, it could have been much better in this
regard.

**Q5 Detailed Comments To The Authors:**

pg 5 "Sepcifically"

top of section 5.2: presumably this should be table 1, not 4

**Q9 Complying With Reviewing Instructions:**

Yes

---

> ### Author Rebuttal · Authors · 2024-04-09
>
> Thank you for your thoughtful review and valuable feedback on our paper. We appreciate your recognition of our paper's strengths, particularly highlighting its well-placed context in related work, the transformer-based architecture for learning in LNS, and the promising experimental outcomes.
>
> Addressing your concerns in detail:
>
> Regarding the handling of variable set decisions and the exploitation of correlations, we acknowledge that the initial description in section 4 lacked clarity. We commit to adding an upfront, comprehensive explanation in the early part of section 4 to emphasize that the correlation exploitation occurs not only during decoding but also at the level of the loss function.
>
> You raised a query about the treatment of large coefficients in real-world instances. We normalize the coefficients by dividing them by the largest coefficient in the instance. We will ensure this crucial step is explicitly mentioned in the manuscript to avoid any ambiguity.
>
> Concerning the encoding into ILP for the four problem families, we followed the same methodology outlined in Wu et al. [NeurIPS 2021]. We will explicitly clarify this in the paper to make our work more transparent.
>
> On the balance of emphasis between randomly generated instances and results on the MIPLIB, we appreciate your feedback and recognize the importance of focusing more on the MIPLIB results. We will restructure the paper to give more prominence to these real-world results, moving them from the supplementary material to the main body where appropriate.
>
> To address reproducibility concerns, we assure you that we have provided sufficient details on dataset generation and model implementation in the paper. Furthermore, upon acceptance, we will release all datasets and source code publicly, ensuring full transparency and facilitating replication of our work.

---

### Official Review · Reviewer_Vr7P · 2024-03-23

**Q2-1 Originality-Novelty:** 3
**Q2-2 Correctness-Technical Quality:** 2
**Q2-5 Clarity Of Writing:** 2

**Q10 Ethical Concerns:**

No, it doesn't raise any new ethical concerns.

**Q1 Summary And Contributions:**

The paper focuses on capturing the sequential correlations between the different iterations of variable selection in the Large Neighborhood Search (LNS). To this end, they propose an algorithm framework, ILP-FORMER, modeling the policy learning as a sequence-to-multi-label classification (MLC) problem, and combine it with an MLC decoder with contrastive learning for variable selection. They also perform experiments to show that it outperforms the state of the art.

**Q2-3 Extent To Which Claims Are Supported By Evidence:**

3: Good: the main claims are supported by convincing evidence (in the form of adequate experimental evaluation, proofs, (pseudo-)code, references, assumptions).

**Q2-4 Reproducibility:**

2: Fair: key resources (e.g. proofs, code, data) are unavailable but key details (e.g. proof sketches, experimental setup) are sufficiently well-described for an expert to confidently reproduce the main results.

**Q3 Main Strengths:**

- The authors compare their algorithms with several baselines and recent work.
- The related work is relatively extensive regarding different aspects of learning algorithms
applied to COPs.
- The algorithm is tested not only on artificially generated benchmarks but also on some
real-world instances.

**Q4 Main Weakness:**

- The presentation is a little difficult to follow.
- The experiment could be made more rigorous (e.g. normalized metrics, more consistent comparison with pastwork)
- The authors claim that their focus on capturing the sequential nature of LNS is what resulted in them outperforming the state of the art. Entangled in the hybrid tools imported from past work, I am not fully convinced about their claim.

**Q5 Detailed Comments To The Authors:**

- the first time when I read Sec4.1, I jumped to the figure and its caption. However, there are some acronyms, such as MLP, which are defined in later paragraphs. They should move the figure afterward. Authors try to boost the LNS framework, so it's better to embed all the explanations or the introduction of the tools in the LNS context. However, the explanation of LNS context is a little weak because the authors use a lot of existing pipelines without justification of the reason: the graph representation [Chen at al., 2021], GNN [Zhang at al., 2019], the causal transformer GPT2 [Radford at al., 2019]...
- Regarding the performance metric (mentioned as "the objective of solutions returned by different algorithms within a time"), they seem to be unnormalized. But you are testing them in a set of instances with different sizes that have an unbalanced influence on overall objective values. Please clarify this clearly.
- More about the comparison experiments, the authors should give the time consumption for all offline algorithms. At least, they should give an overall introduction or comparison. Besides, I am curious about what optimization problems are tested by other comparison algorithms in their original papers. If they are using the same problem set, it is more convincing if the authors' algorithm can outperform them within both the same training and real-solving time. If authors are not using the same problem set, I think it is not convincing to claim, "We [...] fine-tune them on our datasets to get the best hyperparameters." I am curious how you can get the best hyperparameters for algorithms proposed by others. And authors should pay attention to wording in the paper, like "optimal" in the abstract, the "best" here...
- Authors say their Ablation Study in a supplementary file can be justified by mentioning "if we do not consider modeling the sequential process of LNS (drop by 4.09% on average) or exploit correlations of variable selection (drop by 3.13% on average)." But we have to notice that the drop in the percentage of around 3-4% is not enough to convince me if you consider the Std.% to be up to 8.3 in Table 5.
- Considering ILP-FORMER as an offline training paradigm, could the authors also compare the algorithm with recent purely real-time LNS? I am a little curious about the boosting with the sacrifice of long-time training.
- About the iterative calling to Gurobi, the authors gave them 2 seconds to solve sub-ILPs. From my experience, this local timeout is a very important parameter in LNS; trying different values may result in a huge difference. For example, different LNS paradigms have different preferences for the parameter. The fixing of the value may block a lot of information, especially when comparing different algorithms.
- Adding some case analyses can probably help the authors understand what policies their algorithms are learning. For example, they can get some probing features of the selected sub-ILPs or graphs and even compare the local instances selected by different algorithms. Then, they can at least know some intuitions about their blackbox, which integrates a lot of other blackboxes. Or, in their context of time linkage (sequential action), they can analyze whether the selection probability ratio in LIP-FORMER is the same or quite different from the one in the algorithm model without exploiting correlations of variable selection. Which is also a tentative answer of 4).

**Typos**
- "causual" in the abstract
- "un-assigned" on page 2 (should be unassigned)
- "sequance" on page 3
- "bipartitle" and "consistring" on page 4
- "pretrained" several places (should be pre-trained)
- "trahectory" on page 6

**Q9 Complying With Reviewing Instructions:**

Yes

---

> ### Author Rebuttal · Authors · 2024-04-09
>
> Thank you for your insightful review and constructive feedback on our manuscript.
>
> In response to your concerns:
>
> Regarding the clarity of the presentation, we acknowledge that certain aspects might have been challenging to follow and apologize for any confusion caused. We will revise the manuscript to ensure that the descriptions and justifications for using various components, including the graph representation, GNN, and causal transformer GPT2, are embedded more coherently within the context of Large Neighborhood Search (LNS). As per your suggestion, we will clarify that the graph representation follows Gasse et al. [NeurIPS 2019], capturing both static ILP components and dynamic algorithmic states, while GNN is utilized for embedding the graph representation and the causal transformer models the relationships among variable sets in LNS steps.
>
> Concerning the experimental rigor, we appreciate your comment on normalization of performance metrics. We confirm that in each test set, instances have an equal number of variables, as detailed in Section 5.1. Moreover, we ensure that all compared offline algorithms’ time consumption is included in the reported solving time limits, thus providing a fair comparison under the same training and real-solving conditions.
>
> Regarding the ablation study, we understand your point about the significance of the observed performance drop. While we note the standard deviation in Table 5 for the SC-1000 dataset, we maintain that disregarding the sequential process or correlation exploitation in LNS does lead to a noticeable decline in performance. Nonetheless, we will further discuss the implications of these findings in light of the variability present in the data.
>
> On the topic of comparing ILP-FORMER to purely real-time LNS methods, we concur with prior literature (e.g., CL-LNS) that direct comparisons between the two categories are not always equitable due to the inherent advantages of learning-based approaches over purely real-time strategies.
>
> As for the fixed 2-second timeout for solving sub-ILPs using Gurobi, we agree that this is a critical parameter in LNS. Although we adhere to this setup consistent with previous research to ensure a fair comparison, we acknowledge the potential importance of investigating the effects of varying this parameter in future work.
>
> Finally, we appreciate your suggestion for incorporating additional case analyses to gain insights into the learned policies. We will explore options such as analyzing the characteristics of selected sub-ILPs or graphs and comparing local instance selections across different algorithms. This would indeed enhance our understanding of the decision-making processes within our integrated approach, particularly concerning the sequential action context where we could investigate whether exploiting variable selection correlations significantly alters the selection probability ratio in LIP-FORMER.

---

### Meta-Review · Area_Chair_HtJn · 2024-04-16

Although all reviewers (correctly) noted the good empirical performance of the presented method there were (1) some criticism of the (lack of) clarity of the writing and (2) some questions about experimental procedure. Nonetheless, a useful heuristic method for generating primal solutions for ILP problems is given here.